# Ultrafast excited state dynamics and light-switching of [Ru(phen)$_2$(dppz)]$^{2+}$ in G-quadruplex DNA

Chunfan Yang[1], Qian Zhou[1], Zeqing Jiao[1], Hongmei Zhao[2], Chun-Hua Huang[3], Ben-Zhan Zhu[3] & Hongmei Su [1✉]

The triplet metal to ligand charge transfer ($^3$MLCT) luminescence of ruthenium (II) poly-pyridyl complexes offers attractive imaging properties, specifically towards the development of sensitive and structure-specific DNA probes. However, rapidly-deactivating dark state formation may compete with $^3$MLCT luminescence depending on different DNA structures. In this work, by combining femtosecond and nanosecond pump-probe spectroscopy, the $^3$MLCT relaxation dynamics of [Ru(phen)$_2$(dppz)]$^{2+}$ (phen = 1,10-phenanthroline, dppz = dipyridophenazine) in two iconic G-quadruplexes has been scrutinized. The binding modes of stacking of dppz ligand on the terminal G-quartet fully and partially are clearly identified based on the biexponential decay dynamics of the $^3$MLCT luminescence at 620 nm. Interestingly, the inhibited dark state channel in ds-DNA is open in G-quadruplex, featuring an ultrafast picosecond depopulation process from $^3$MLCT to a dark state. The dark state formation rates are found to be sensitive to the content of water molecules in local G-quadruplex structures, indicating different patterns of bound water. The unique excited state dynamics of [Ru(phen)$_2$(dppz)]$^{2+}$ in G-quadruplex is deciphered, providing mechanistic basis for the rational design of photoactive ruthenium metal complexes in biological applications.

[1] College of Chemistry, Beijing Normal University Institution No.19, Haidian District, Beijing, China. [2] Institute of Chemistry, Chinese Academy of Sciences, Beijing, China. [3] State Key Lab of Environmental Chemistry and Ecotoxicology, Research Center for Eco-Environmental Science Chinese Academy of Sciences, Beijing, China. ✉email: hongmei@bnu.edu.cn

Among the transition metal complexes, the luminescent ruthenium (II) polypyridyl complexes have attracted great research interests owing to their unique photophysical and photochemical properties[1–5]. [Ru(phen)$_2$(dppz)]$^{2+}$ has been well-known as a benchmark luminescent DNA marker for its molecular "light-switch" effect[6–9]. Upon photoexcitation in visible, the metal-to-ligand charge transfer ($^1$MLCT) state of [Ru(phen)$_2$(dppz)]$^{2+}$ relaxes to a triplet state ($^3$MLCT) within ~300 fs, which is a "bright state" and should emit red luminescence. However, the luminescence is extinguished in water because $^3$MLCT decays to a "dark state" within ~3 ps by the formation of hydrogen bonds between the phenazine (phz) nitrogen atoms of the dppz ligand and solvent water[10]. This process can be well described with the kinetic model in Fig. 1. Remarkably, when [Ru(phen)$_2$(dppz)]$^{2+}$ binds with double-stranded DNA (ds-DNA) the luminescence is drastically switched on, for the dppz ligand is intercalated between base-pair steps that afford hydrophobic environment preventing protonic water access. Generally, the "light-switch" effect shows that the ruthenium complexes are protected from the aqueous environment, and the changes of luminescent emissions upon binding with nucleic acid provide a convenient method to evaluate the extent of interaction[11].

Owing to their sensibility to the different microenvironments and notable luminescent emissions, [Ru(phen)$_2$(dppz)]$^{2+}$ and its derivatives offer a very attractive set of optical imaging properties, specifically towards the development of highly sensitive and structure-specific DNA probes[12,13]. Interestingly, a dinuclear Ru (II) complex [(phen)$_2$Ru(tpphz)Ru(phen)$_2$]$^{4+}$ has been successfully applied in cellulo nuclear DNA stain[14]. The dinuclear Ru (II) complex was found to display a distinctive blue-shifted 'light-switch emission on its binding to G-quadruplex DNA compared with the duplex DNA (maxima of ~630 and >650 nm, respectively), which endowed ruthenium complexes a promising probe for recognizing different structures of DNA. The underlying molecular mechanisms for the "light-switch" emission in G-quadruplex distinct from duplex DNA could be associated with the local microenvironment that these structures afford. However, so far in contrast to the studies for duplex DNA, much less is known for the interplay of ruthenium complex binding with noncanonical DNA motifs such as G-quadruplex.

G-quadruplex is a family of stable four-stranded structures formed by folding of certain guanine-rich sequence, with stacking of planar G-quartets comprising of four guanines via Hoogsteen hydrogen bonds. Research into G-quadruplexes has aroused a strong attention[15–20] because these structures are prevalent in genomes and act as regulators in key biological processes of DNA replication, transcriptions and damage/repair. Structurally, G-quadruplexes manifest complex shapes rather than being purely linear polymeric assemblies[21], consisting of a variety of core and loop elements with different hydrogen bonding, solvent accessibility, and base stacking. Previous crystallography and NMR studies[22,23] indicated that water molecules can link the bases and the coordinate ion of K$^+$ at the centre of the loop as bridges, where water molecules play essential roles for stabilizing the loop conformation and the G-quadruplex topology. The non-uniform hydration pattern with increasing complexity in G-quadruplex, thus raises the critical question of the local excited state dynamics of $^3$MLCT for Ru (II) polypyridyl complexes within the microenvironment of the G-quadruplex. Virtually, this question is the core foundation for the rational design of Ru (II) photoprobes and for understanding the significant roles of "biological water" in changing excited state decay pathways[24].

In this work, we utilize [Ru(phen)$_2$dppz]$^{2+}$ as environmentally sensitive photoprobes for visualization of the microenvironment hydration properties of G-quadruplexes and to probe the local hydration effect on excited state dynamics. The light-switch and photodynamics of [Ru(phen)$_2$dppz]$^{2+}$ in two iconic G-quadruplexes with well-defined structures (the *HumanTelomere* (HT) and *Oxyticha nova*), compared with in bulk water and in ds-DNA, are systematically examined by combining femtosecond and nanosecond pump-probe spectroscopy. It is found that the inhibited dark state channel in ds-DNA is open in G-quadruplex, and the ultrafast dark state formation dynamics of [Ru(phen)$_2$dppz]$^{2+}$ is sensitive to the content of water molecules in local DNA structures. The dark state formation rate of [Ru(phen)$_2$dppz]$^{2+}$ is ~10.5 ps when bound to the bilateral TTA (thymine-thymine-adenine) loops of HT and is ~15.6 ps when bound to the diagonal thymine–thymine–thymine–thymine (TTTT) loops of *Oxyticha nova*, indicating different patterns of bound water molecules contained in different structural loops. The small amount of local water within G-quadruplex thus play key roles in depopulating [Ru(phen)$_2$dppz]$^{2+}$ molecules from $^3$MLCT state to a dark state, resulting in unique local excited state dynamics different from ds-DNA. We note that the photophysical properties of [Ru(phen)$_2$(dppz)]$^{2+}$ binding with G-quadruplex is allowed to be characterized by ultrafast time-resolved spectroscopy from dynamics perspective, which is of fundamental importance for the rational design of photoactive ruthenium metal complexes in nucleic acid research and cancer cell imaging.

## Results and discussion

**Steady-state and nanosecond time-resolved spectroscopy characterizing light-switch and binding mode.** The absorption spectra of [Ru(phen)$_2$dppz]$^{2+}$ alone and in the presence of G-quadruplexes are depicted in Fig. 2a. The absorption bands of [Ru(phen)$_2$dppz]$^{2+}$ in the visible region are attributed to the MLCT transitions (439 nm) and the intraligand transition (372 nm), respectively. After being added to HT G-quadruplex, the absorption of [Ru(phen)$_2$dppz]$^{2+}$ exhibits hypochromism and red shift, suggesting the binding with G-quadruplex. Furthermore, the binding constant ($K = 5 \times 10^6 \, \text{M}^{-1}$) can be determined according to the UV absorption change (Fig. 2b) based on reported methods[13,25] (see Supplementary Methods for Absorption spectra titration). This binding constant is comparable to that for these complexes with ds-DNA (~$10^6 \, \text{M}^{-1}$)[26], and similar to the binding constant between [Ru(bpy)$_2$dppz]$^{2+}$ and G-quadruplex[27], indicating that [Ru(phen)$_2$dppz]$^{2+}$ can also bind strongly with G-quadruplex. Known to have "light-switch" effect, [Ru(phen)$_2$dppz]$^{2+}$ shows negligible emission in water, while the emission intensities can be dramatically enhanced when binding to DNA. As shown in Fig. 2c, the emission at 620 nm from the $^3$MLCT of [Ru(phen)$_2$dppz]$^{2+}$ is greatly enhanced when binding

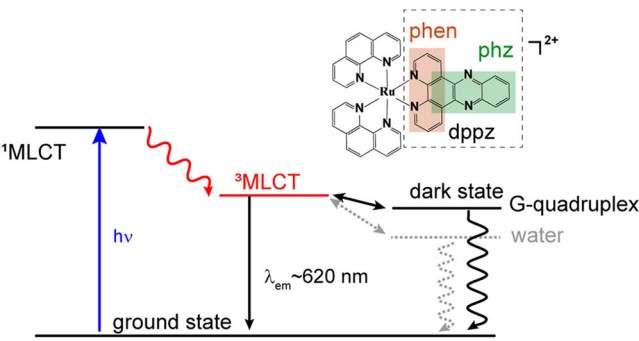

**Fig. 1 Jablonski diagrams for the photophysical processes of [Ru (phen)$_2$(dppz)]$^{2+}$.** The phenanthrolin (phen) and phenazine (phz) moieties of dppz ligand are highlighted in the molecular structure.

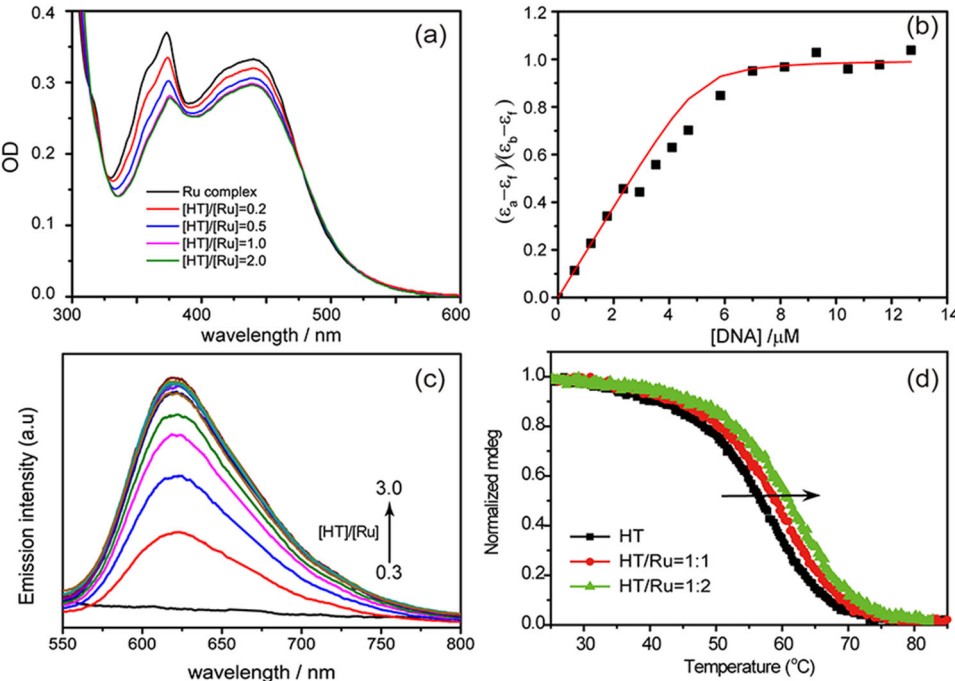

**Fig. 2 Interactions between [Ru(phen)₂dppz]²⁺ and G-quadruplex measured by the steady-state methods. a** UV–vis absorption spectra of 5 µM [Ru(phen)₂dppz]²⁺ alone and when bound to HT G-quadruplex in 10 mM Tris–HCl, and 100 mM NaCl buffer (pH 7.5) at several concentration ratios. **b** Determination of the binding constant ($K$) by measuring $(\varepsilon_a-\varepsilon_f)/(\varepsilon_b-\varepsilon_f)$ as a function of DNA concentration, where $\varepsilon_a$, $\varepsilon_f$, and $\varepsilon_b$ are, the apparent extinction coefficient ($A/[M]$), the extinction coefficient for free metal complex $M$ and the extinction coefficient for $M$ in the fully bound form respectively. **c** The emission spectra of [Ru(phen)₂dppz]²⁺ alone and when bound to HT G-quadruplex at several concentration ratios (**d**) Normalized CD melting curves for HT G-quadruplex in the absence and presence of [Ru(phen)₂dppz]²⁺. The stability of G-quadruplexes was assessed by the CD signal at 295 nm (Supplementary Fig. 1).

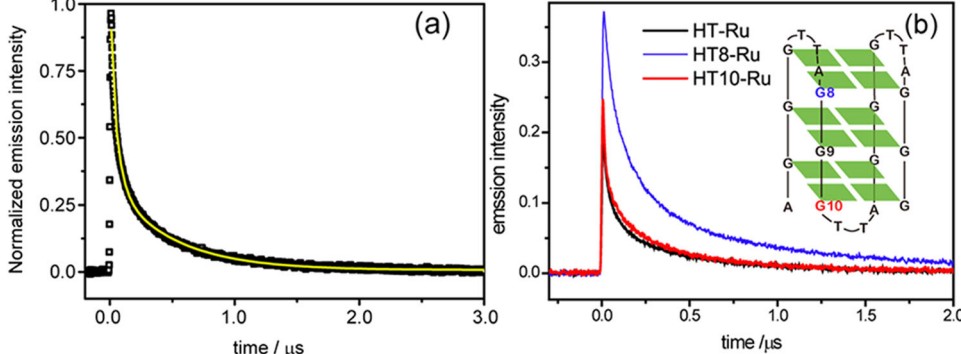

**Fig. 3 The photodynamics of [Ru(phen)₂(dppz)]²⁺ with G-quadruplex in nanosecond time scale. a** Experimental (black) and fitted (yellow) decay dynamics curves of ³MLCT luminescence at 620 nm for [Ru (phen)₂dppz]²⁺ when bound to HT G-quadruplex in 10 mM Tris–HCl, and 100 mM NaCl buffer (pH 7.5) upon 355 nm excitation. The same experiments were obtained upon 400 nm excitation. **b** Normalized ³MLCT luminescence decay dynamics curves upon 355 nm excitation for the [Ru(phen)₂dppz]²⁺ when bound to normal HT AG₃(T₂AG₃)₃ and mismatch G-quadruplexes HT8 and HT10, the inset figure shows the numbering of G base substituted by T base in HT G-quadruplex.

with HT G-quadruplex, indicating the pronounced "light-switch" effect for [Ru(phen)₂dppz]²⁺/G-quadruplex.

The HT G-quadruplex adopts antiparallel basket-type conformation in the presence of Na⁺ ions, as confirmed by the features of a positive peak at 295 nm and a negative peak at 263 nm in the circular dichroism (CD) spectra (Supplementary Fig. 1)[18]. Meanwhile, thermal CD experiments (Fig. 2d) show the increasing melting temperature for HT G-quadruplex, $T_m$ (56.6, 58.8, 61.0 °C) with the increasing concentration of [Ru(phen)₂dppz]²⁺, indicating that the G-quadruplex structure can be stabilized by [Ru(phen)₂dppz]²⁺. This first rules out the possibility of intercalation binding mode, since intercalation

between two G-quartets tends to destabilize the G-quadruplex structure[28,29].

To determine the binding mode, nanosecond time-resolved luminescence dynamics was measured. As shown in Fig. 3a, the ³MLCT emission at 620 nm for [Ru(phen)₂(dppz)]²⁺ tethered to HT G-quadruplex displays a biexponential decay dynamics ($y = Ae^{-1/\tau L} + Be^{-t/\tau S}$), with two lifetime components of $\tau_L = 490.7$ ns and $\tau_S = 57.9$ ns due to the local bound microenvironment that prevents the bulk water quenching. The observation of two lifetimes should be related to the extent of protection of the bound local environment. The more exposed to bulk water, the larger the polarity of the microenvironment, and the shorter the

**Table 1 The $^3$MLCT luminescence lifetimes of [Ru(phen)$_2$dppz]$^{2+}$ in DNA, obtained from bi-exponential fitting, with preexponential factors for the two lifetime components shown in parentheses.**

| Ruthenium complex with | $\tau_L$ (ns) | $\tau_S$ (ns) |
|---|---|---|
| HT | 490.7 ± 3.2 (0.26) | 57.9 ± 0.7 (0.74) |
| HT8 | 1046.0 ± 9.7 (0.29) | 151.2 ± 1.2 (0.71) |
| HT10 | 430.7 ± 4.4 (0.57) | 61.4 ± 1.5 (0.43) |
| dsDNA | 492 ± 1.7 (0.23) | 88.1 ± 0.2 (0.77) |
| *Oxyticha nova* | 124.1 ± 2.0 (0.07) | 23.0 ± 0.1 (0.93) |

The error bar is given based on the fitting uncertainties.

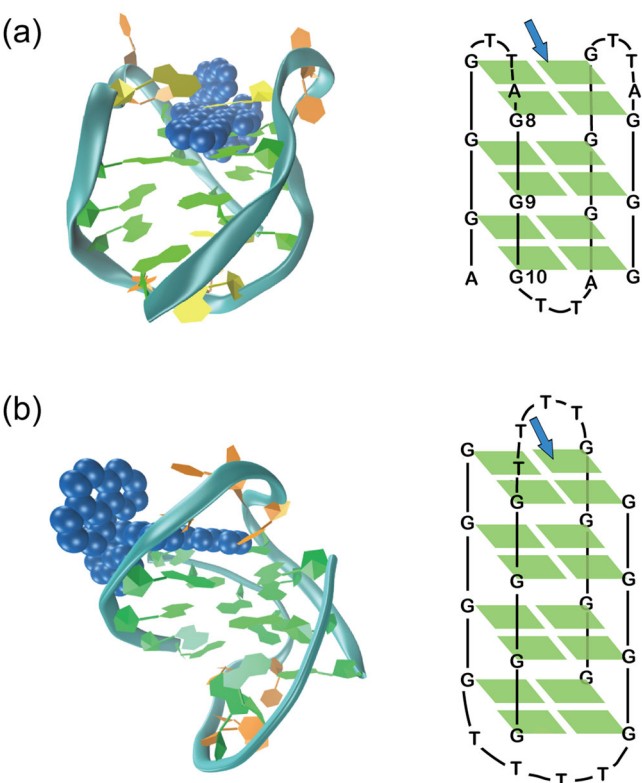

**Fig. 4 Schematic structures of [Ru(phen)$_2$dppz]$^{2+}$ with two G quadruplexes.** The sketch of [Ru(phen)$_2$dppz]$^{2+}$ bound to (**a**) *human telomere* G quadruplex by stacking of dppz on the TTA bilateral loop end, and (**b**) *Oxyticha nova* G-quadruplex by stacking of dppz on the TTTT diagonal loop end. The ruthenium complex is shown in blue. Guanine, thymine and adenine residues are shown in green, orange and yellow, respectively. Schematic structures of the two G quadruplexes are also depicted with the binding sites indicated by blue arrows. The G-quadruplex structure based on NMR measurements (PDB 143D) is used here.

luminescence lifetime[10]. [Ru (phen)$_2$dppz]$^{2+}$ binding with the ds-DNA formed by HT sequence with its complementary sequence was also measured for comparison. As shown in Table 1, the $^3$MLCT luminescence lifetimes in HT G-quadruplex are comparable to those in ds-DNA ($\tau_L = 492$ ns and $\tau_S = 88.1$ ns), indicating that the depth of dppz ligand approaching into HT G-quadruplex core should be similar to its counterpart ds-DNA. The $^3$MLCT luminescence lifetimes of [Ru(phen)$_2$dppz]$^{2+}$ in the current DNA systems are both shorter than that in pure nonaqueous solvent, which was reported to be 663 ns in CH$_3$CN[7].

Stacking on the terminal G-quartets (end-stacking) is usually a most likely binding mode with G-quadruplex, given that this interaction can readily occur through π–π stacking and is unhindered by the need to change lateral and diagonal loop conformation to do so[30]. The cationic [Ru(phen)$_2$dppz]$^{2+}$ possesses a more extended heteroaromatic ligand dppz that is favourable for π-stacking with terminal G-quartet with a high binding constants ($K = 5 \times 10^6$ M$^{-1}$), as schematically shown in Fig. 4. In Na$^+$ solutions, the human telomeric sequence d[AG$_3$(T$_2$AG$_3$)$_3$] adopts an antiparallel stranded basket-type topology: two TTA lateral loops are involved with one terminal G-quartet, and a TTA diagonal loop with the other terminal[31]. These TTA loops extend laterally from the core G-quartets (Fig. 4a), providing cavities which could accommodate ruthenium (II) complex. Therefore, the terminal G-quartets with lateral or diagonal loops are both possible interfaces for the interactions between [Ru(phen)$_2$dppz]$^{2+}$ and HT G-quadruplex DNA. In this case, are the two lifetimes of the $^3$MLCT emission attributed to these two binding sites, respectively?

To distinguish further the binding site, control experiments were performed using the mismatch *human telomere* sequence to interact with [Ru(phen)$_2$dppz]$^{2+}$. Specific guanine base (G8, G9, G10) is replaced by thymine base, respectively, and labelled as HT8, HT9, HT10 (shown in Fig. 3b inset). It was reported that substitutions of G by T in the terminal quartets did not change the wild-type's antiparallel fold in Na$^+$ buffer[32]. According to the CD spectra (Supplementary Fig. 1), HT8 and HT10 retain the antiparallel configurations while HT9 totally changes, indicating that the integrity of the core G9-quartet is the key to sustain the overall structure. The G9 mismatch (HT9) caused the disruption of G-quadruplex structure. Thus, we monitor the emission decay kinetics of [Ru(phen)$_2$dppz]$^{2+}$ binding with HT8 and HT10 that retain the G-quadruplex integrity. Unlike guanine, thymine has only one heterocyclic ring, therefore T base forms less and weaker Hoogsteen hydrogen bonds with the neighbouring G bases, forming a mismatched TGGG quartet. As a result, the replacement of T base (mismatch) tends to weaken the π–π stacking of G-quartets[33] and such changes should be helpful for the ruthenium complexes approaching G-quadruplex core, which will offer better protection from the water, it is expected that the emission lifetime of [Ru(phen)$_2$dppz]$^{2+}$ should be increased if the mismatch point is the binding site. Interestingly, it is observed that (Fig. 3b), compared to the normal HT G-quadruplex, the emission lifetimes increase greatly in HT8 ($\tau_L = 1046$ ns, $\tau_S = 151$ ns), whereas the emission lifetimes in HT10 do not change much (Table 1). Accordingly, the emission intensity of [Ru(phen)$_2$dppz]$^{2+}$ binding with HT8 is obviously enhanced relative to HT, whereas HT10 is about the same as HT. These observations indicate that [Ru(phen)$_2$dppz]$^{2+}$ mainly stacks on the terminal of the G8-quartet with the lateral loops, while the percentage of binding with the diagonal loop should be small, although both terminal quartets are potential binding sites[34]. Our results consist with the previous molecular docking calculations, which predicted the terminal of the HT G-quadruplex with bilateral loops to be the favourable binding site[13,35]. The bilateral loop cavity should have larger flexibility to accommodate the non-planar [Ru (phen)$_2$dppz]$^{2+}$ more easily than the rigid diagonal loop cavity, resulting into the preferential binding on the bilateral terminal (Fig. 4a). Since the ruthenium complexes mainly stacks on the G8-quartet, the replacement of G10 by T does not lead to obviously different time constants in HT10 compared to HT. The change of the preexponential factors in HT10 should be related to the small perturbation of the whole G-quadruplex structure upon G10 mismatch. This small perturbation might be beneficial for the ruthenium complexes to approach

into the core of the G-quadruplex, which is implied by the increased preexponential factor of the long lifetime $\tau_L$.

Meanwhile, the two emission lifetimes of $[Ru(phen)_2dppz]^{2+}$ ($\tau_L$ and $\tau_S$) should correspond to two orientations of terminal stacking. Stacking fully ($\tau_L = 490.7$ ns) on the G8-quartet can render phenazine nitrogens relatively well shielded by the quartet and loops, whereas stacking partially ($\tau_S = 57.9$ ns) may leave partial exposure to water quenching. These assignments resemble the case in double strand DNA, and the two lifetimes assigned to two binding orientations has been well established in ds-DNA. For instance, for $[Ru(phen)_2dppz]^{2+}$ with calf thymus DNA[36], the long lifetime (750 ns) was ascribed to a fully intercalated binding of dppz ligand within hydrophobic base-pair steps of the double helix, and the short lifetime (120 ns) was attributed to the partial intercalation with a more canted orientation and more exposure to water. In addition, $[Ru(phen)_2dppz]^{2+}$ has two enantiomers which both bind extremely strongly to ds-DNA without any noticeable enantioselectivity. The $\Delta$ enantiomer showed stronger emissions upon ds-DNA binding, and the reason was ascribed to a slightly different intercalation geometry which resulted in a different degree of protection from quenching by water. Each of the enantiomers show biexponential luminescence[37] decay which can be attributed to full and partial stacking. The two lifetimes of $[Ru(phen)_2dppz]^{2+}$ with calf thymus DNA were still assigned to the different binding mode of the racemate[36] instead of to the different enantiomers. In analogy to the case in ds-DNA, both $\Lambda$ and $\Delta$ enantiomers strongly bind with the G-quadruplex. The binding constant of the derivates of $[Ru(phen)_2dppz]^{2+}$ with HT G-quadruplex are $2.7 \times 10^6$ M$^{-1}$ for $\Delta$ and $4.6 \times 10^6$ M$^{-1}$ for $\Lambda$ enantiomers, respectively[38]. The reported stronger emission of $\Lambda$ enantiomer upon binding with HT G-quadruplex should be also due to the slightly different binding geometry[39].

**Ultrafast photodynamics of $[Ru(phen)_2dppz]^{2+}$ in water, ds-DNA, and G-quadruplex DNA.** After ascertaining the binding sites, the ultrafast photodynamics of $[Ru(phen)_2dppz]^{2+}$ in G-quadruplexes were monitored by femtosecond transient absorption spectroscopy, in comparison with that in water and in ds-DNA. As is well-known, $[Ru(phen)_2dppz]^{2+}$ is quite stable photochemically, because its excited state is insufficiently oxidizing to cause the electron transfer with guanine[40]. Therefore, there are only photophysical decay pathways involved in the current ds-quadruplex and G-quadruplex DNA systems although they are G-rich. The transient absorption spectra were recorded in 0–1000 ps time range after excitation with a 100-fs pulse of 400 nm laser. The pump wavelength 400 nm is selected, which falls at the blue edge of the MLCT absorption band of the ruthenium complex, for obtaining a broad range (>400 nm) of fs-transient absorption spectral detection covering the signal of ground state depletion. The fs transient-absorption spectroscopy in principle samples the depletion of the ground state (negative signal) and formation of all accessible excited states (positive signal) including the long-lived emissive state ($^3$MLCT) and the short-lived dark state, which allows a holistic picture of the photodynamics to be revealed.

In aqueous solution, it is known that following the initial photoexcitation, the $^1$MLCT state of ruthenium complex relaxes to a triplet state ($^3$MLCT) within ~300 fs[41,42], which then undergoes deactivation to a "dark state" with a rate of ~3 ps due to the hydrogen bonding with the phenazine nitrogens by the ubiquitous water in solution[10,43]. The dark state decays to the ground state with a time constant of ~250 ps. Accordingly, in the transient absorption spectra (Fig. 5a) the excited state absorption (ESA) bands of $^3$MLCT state peaking at 600 nm are observed

initially, with its decay in the first 1 ps, there is a wavelength shift and build-up of the dark state band at 550 nm, which then fully decays to baseline within 1000 ps. Concomitantly, the ground state bleach (GSB) band at 440 nm shows very little recovery during the first 1 ps, and then recovers gradually back to the baseline within 1000 ps. It is also noticeable that in addition to the peak wavelength shift from 600 to 550 nm, there is a decrease of intensity for the broad and flat absorption ~700 nm, when $^3$MLCT state converts to dark state for $[Ru(phen)_2(dppz)]^{2+}$ in bulk water. A global multiexponential fit of the data set yields two time constants of 2.4 and 248 ps for the interconversion from $^3$MLCT state to dark state and dark state decay to ground state, respectively. These time constants agree well with literature[10]. The corresponding evolution-associated difference spectra (EADS) spectra obtained from global fit is shown in Fig. 6a, which matches the observed spectral shape and provides rationale for the global fit. Additionally, the fitted spectra matches the experimental data (shown in Supplementary Figs. 8–10), which also proves the accuracy of the global fit.

When $[Ru(phen)_2dppz]^{2+}$ binds with the ds-DNA (formed by HT sequence with its complimentary strand), as shown in Fig. 5b, there is no recovery of the ground state, and the $^3$MLCT absorptions spanning 500 nm to above 700 nm remain constant up to the instrumental time window of 1 ns. Hence the relaxation of $^3$MLCT to the dark state does not occur. Instead, the lifetime of $^3$MLCT ($\tau_{em}$) is on the hundreds of nanoseconds time scale, as shown in our nanosecond emission experiments, and the radiative relaxations become predominant. Obviously, the long-lived $^3$MLCT state serves as a reservoir for the excited state populations, which makes the ground state depletion band at ~440 nm subject to very little recovery up to the 1 ns time window. This observation agrees with previous time-resolved infrared spectroscopy studies, showing that the decay of $^3$MLCT to the dark state is highly suppressed for $[Ru(phen)_2dppz]^{2+}$ bound to ds-DNA. The intercalation of $[Ru(phen)_2dppz]^{2+}$ within the hydrophobic base-pair steps protects the dppz ligand from the water molecules, and thus blocks the interconversion to the dark states[44,45].

Previous investigations[36] have addressed clearly that there are two binding orientation in ds-DNA, the full intercalation with both the phenazine nitrogen atoms being protected, and the partial intercalation with one of the phenazine nitrogen atoms being protected and the other being partially exposed in the DNA major groove. Although the partial intercalation seems to offer less protection for the dppz ligand and could allow partial exposure to water, it is noteworthy that the dark state channel of $[Ru(phen)_2dppz]^{2+}$ is still totally inhibited in ds-DNA. These results indicate that the hydration feature of local bound microenvironment should be the determinant for the dark state formation. For ds-DNA, the base-pairs steps provide hydrophobic surroundings for $[Ru(phen)_2dppz]^{2+}$, whether it adopts full or partial intercalation, such that hydrogen bonding and dark state formation is completely blocked.

Interestingly, the fs transient absorption spectra for $[Ru(phen)_2dppz]^{2+}$ in HT G-quadruplex is clearly different from that in ds-DNA. As shown in Fig. 5c, the temporal evolution of the fs TA spectra displays a partial recovery of GSB (440 nm), in concomitant with the decay of flat absorption ~700 nm and the discernible build-up around 600 nm. These are the features of dark state formation, which are superimposed on the top of the overall transient absorption offset (constant up to 1 ns). The offset corresponds to the long-lived emissive state, $^3$MLCT, which has hundreds of ns lifetime and acts as a reservoir. These transient spectral features of $[Ru(phen)_2dppz]^{2+}$ in HT G-quadruplex indicate that most $^3$MLCT populations still undergo

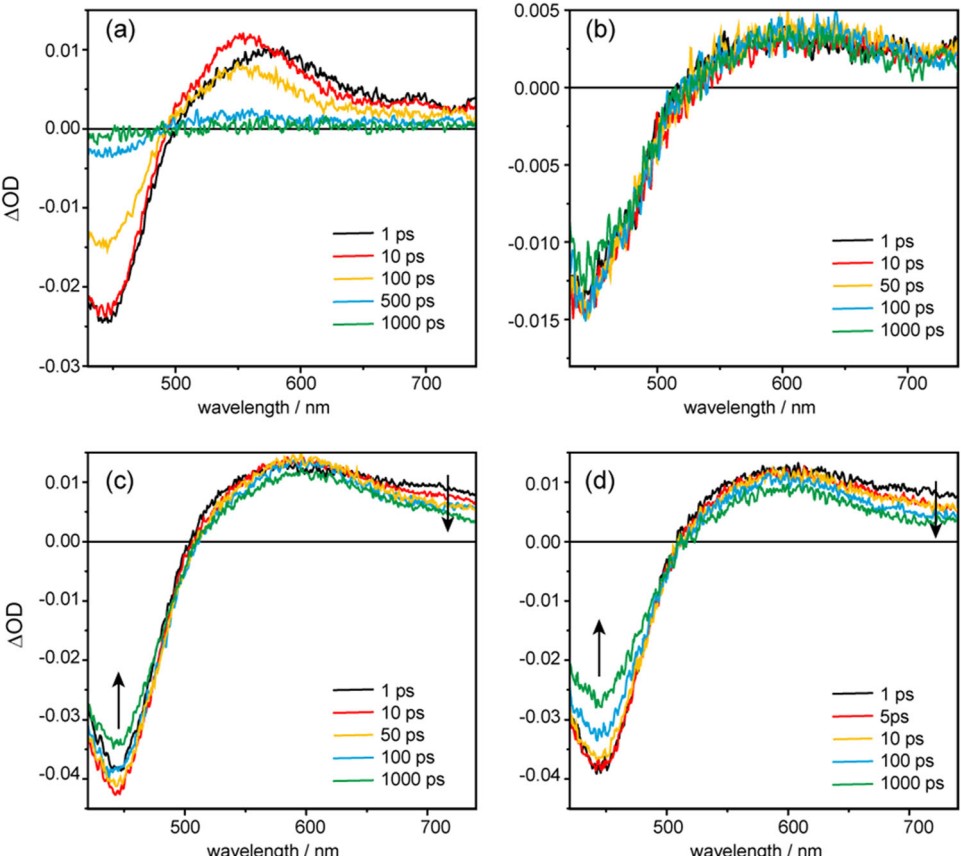

**Fig. 5 The ultrafast photodynamics of [Ru(phen)₂dppz]²⁺ in G-quadruplexes monitored by femtosecond transient absorption spectroscopy.** Ultrafast transient absorption spectra of [Ru(phen)₂dppz]²⁺ upon 400 nm fs laser excitation in water (**a**), ds-DNA (**b**), HT G-quadruplex (**c**), and *Oxyticha nova* G-quadruplex (**d**) in 10 mM Tris–HCl, and 100 mM NaCl buffer (pH 7.5). The concentrations used are [Ru(phen)₂(dppz)]²⁺ = 66 μM, [DNA] = 132 μM. The change of excitation wavelength to 350 nm does not change the transient absorption spectra (Supplementary Fig. 11). Though 350 and 400 nm excitation may change the initial distribution of the excited states, the ultrafast depopulation results in the same ¹MLCT state and thus the same processes being observed.

emissive decay, while only a partial ³MLCT population is branched to the dark state relaxation channel.

The fs transient absorption data are quantitatively analysed by a global fit, from which the ultrafast relaxation dynamics of dark state channel can be characterized. The best fit was achieved by applying a sum of three exponentials, representing three relaxation processes (Fig. 1), and an offset to model the long-lived ³MCLT state, which does not decay on the time scales accessible with the femtosecond setup (1000 ps). The characteristic time constants of the involved processes are $\tau_1 = 2.7$ ps, $\tau_2 = 10.5$ ps, and $\tau_3 = 482$ ps, and the corresponding EADS spectra are displayed in Fig. 6b. Given that the intersystem crossing of [Ru(phen)₂dppz]²⁺ from ¹MLCT to ³MLCT is ~300 fs, the first time constant ($\tau_1 = 2.7$ ps) should be attributed to subsequent "evolution" of the ³MLCT state to a stable quasi-equilibrium via solvent reorganization or vibrational cooling, consistent with reports for related Ru complexes[46]. The solvent reorganization process is usually slower in DNA than that in water because of DNA structure restriction. The other two-time constants correspond to the processes of ³MLCT state relaxation to the dark state ($\tau_2 = 10.5$ ps) and the dark state quench to ground state ($\tau_3 = 482$ ps), respectively.

It shows here that the dark state formation channel is open for [Ru(phen)₂dppz]²⁺ in G-quadruplex. The distinct dark state formation dynamics should be related to the local microenvironment of G-quadruplex where the ligand binds. According to our ns-luminescence dynamics data, the terminal G-quartet with bilateral loops is the principal binding sites, where the large aromatic dppz ligand can achieve good overlap with the G-quartet plane and gain protection for one side of phenazine nitrogen atoms. Meanwhile, the other side of phenazine nitrogen atoms are still exposed to the bilateral TTA loop cavity. The TTA bases can offer some protection (as shown in Fig. 4), however, it is expected that the bilateral TTA loops have flexible conformation, which can open from time to time and allow the dynamic entering of water molecules. Consequently, some mobile water molecules could be present in the bilateral loop cavity, resulting into hydrogen bonds with the dppz ligand and the opening of the dark state channel for [Ru(phen)₂dppz]²⁺ bound to G-quadruplex. This is different from the case in ds-DNA, where the base-pair microenvironment is mostly hydrophobic such that the dark state channel is completely inhibited. On the other hand, the dark state formation rate in G-quadruplex (10.5 ps) observed here for [Ru(phen)₂dppz]²⁺ is much slower than that in bulk water (2.7 ps), but is comparable to that measured in mixed solvents of 1:2 CH₃CN/H₂O (10 ps)[10]. This suggests that the local bound microenvironment in G-quadruplex behaves as a partially hydrophobic and partially hydrophilic solvent resembling the mixed solvent of 1:2 CH₃CN/H₂O. This interesting hydration feature for HT G-quadruplex is thus revealed here by ultrafast photodynamics studies. It implicates that the flexible opening of the bilateral loops may retain dynamic entering water molecules, causing a more hydrophilic microenvironment.

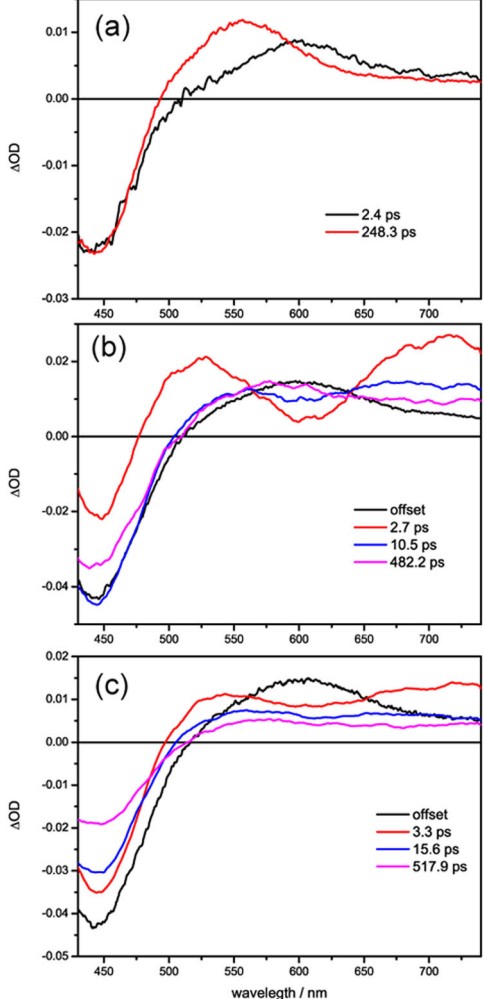

**Fig. 6 The ultrafast transient absorption data quantitatively analyzed by a global fit.** Evolution-associated difference spectra (EADS) of the corresponding kinetics components obtained from global analysis of the ultrafast transient absorption spectra for $[Ru(phen)_2dppz]^{2+}$ in water (**a**), HT G-quadruplex (**b**), and *Oxyticha nova* G-quadruplex (**c**).

**Table 2 Time constants for ultrafast excited state decay processes of $[Ru(phen)_2dppz]^{2+}$ in water and in G-quadruplexes, obtained from global fit of fs transient absorption data.**

| Ru-complex in | $\tau_1$ (ps) | $\tau_2$ (ps) | $\tau_3$ (ps) |
|---|---|---|---|
| Water | 0.3[a] | 2.4 ± 0.03 | 248 ± 0.7 |
| Human telomere | 2.7 ± 0.1 | 10.5 ± 0.3 | 482 ± 5 |
| *Oxyticha nova* | 3.3 ± 0.1 | 15.6 ± 0.2 | 517 ± 8 |

The error bar is given based on the fitting uncertainties.
[a]The ultrafast intersystem crossing from $^1$MLCT to $^3$MLCT of $[Ru(phen)_2dppz]^{2+}$ is beyond our experimental time resolution, so the time constant 0.3 ps reported for a similar compound $[Ru(bpy)_3]^{2+}$ from refs. [37,38] is listed here.

dark state relaxation channel is solely caused by the local bound microenvironment of the G-quadruplex, and vice versa, the dark state formation rate is an indicator of the hydration pattern of the local G-quadruplex structure, which is distinct from bulk water and also distinct from ds-DNA.

To delve further, the fs transient absorption spectroscopy experiments were performed for $[Ru(phen)_2dppz]^{2+}$ binding with *Oxyticha nova* G-quadruplex, which is a typical structure with diagonal topology (Fig. 4b) as confirmed by CD spectra (Supplementary Fig. 2). The *Oxyticha nova* G-quadruplex is composed of two antiparallel hairpins, with four stacked quartets and two identical TTTT diagonal loops at the ends[22,47]. $[Ru(phen)_2dppz]^{2+}$ can also bind strongly with *Oxyticha nova* G-quadruplex, with a binding constant of $1 \times 10^6 M^{-1}$ (Supplementary Table 1) and a substantial light-switch effect (Supplementary Fig. 5). Thermal CD experiments (Supplementary Fig. 3) first rule out the intercalation binding between two G-quartets, as indicated by the no change of melting temperature 59.2 °C upon binding with $[Ru(phen)_2dppz]^{2+}$. Unlike the TTA diagonal loop in HT G-quadruplex, the TTTT loop of *Oxyticha nova* is larger and thus offers larger room for accommodating $[Ru(phen)_2dppz]^{2+}$. Presumably, $[Ru(phen)_2dppz]^{2+}$ still adopts the end-stacking binding mode (Fig. 4b), by stacking the aromatic dppz ligand (fully and partially) onto the terminal G-quartet at either TTTT ends. Since the TTTT diagonal loop region of *Oxyticha nova* provides protections for the Ru complex different from the bilateral TTA loop region of HT G-quadruplex, and the binding constants for these two G-quadruplex are different, the emission lifetimes of $Ru(phen)_2dppz]^{2+}$ with *Oxyticha nova* G-quadruplex are different from that of $Ru(phen)_2dppz]^{2+}$ with HT G-quadruplex (Supplementary Fig. 6 and Table 1), which is consistent with the steady-state emission intensities of $[Ru(phen)_2dppz]^{2+}$ (Supplementary Fig. 7). In the observed fs transient absorption spectra (Fig. 5d) for $[Ru(phen)_2dppz]^{2+}$/*Oxyticha nova* G-quadruplex, there is the feature for $^3$MCLT transformation to dark state, as manifested by the temporal evolution of the decreased ESA intensity around 700 nm and the partial recovery of GSB at 440 nm. Global fit with three exponentials and the long-lived offset gives time constants for three processes (Fig. 6c, Table 2): the structural and electronic equilibration to $^3$MCLT state ($\tau_1 = 3.3$ ps), the $^3$MCLT decay to dark state ($\tau_2 = 15.6$ ps), and the dark state quench to ground state ($\tau_3 = 517$ ps). Obviously, the dark state formation rate (15.6 ps) is slowed down compared to that in HT G-quadruplex (10.5 ps).

On the other hand, despite the seemingly large protection in the TTTT loop region for the dppz ligand that prevents water access, there is still a large population branching to the dark state relaxation channel, as shown by the modest GSB recovery (Fig. 5d). Interestingly, previous crystallography studies[22,47] indicated a pronounced pattern of hydration in the *Oxyticha nova* G-quadruplex, where cations (Na$^+$ or K$^+$) located between

Hypochromic effect is usually seen in the absorption spectra (including fs-TA) of a ligand upon binding to DNA. Moreover, the local microenvironment of G-quadruplex where $[Ru(phen)_2dppz]^{2+}$ binds is only partially hydrophilic, which makes the lowering of dark state energy relative to $^3$MLCT state less than that in bulk water (Fig. 1 and detailed discussion later). Therefore, it shows that the ESA band of dark state does not change much compared to the ESA band of $^3$MLCT state, both of which fall at around 600 nm (Fig. 5c). This explains why the temporal evolutions of $[Ru(phen)_2dppz]^{2+}$ in G-quadruplex in fs-TA spectra are not as obvious as that in bulk water. This phenomena in turn demonstrates the distinct effect of G-quadruplex local environment on the dark state formation dynamics.

In the above system, almost all $[Ru(phen)_2dppz]^{2+}$ molecules are bound to HT G-quadruplex and there is negligible (<2%) free $Ru(phen)_2dppz]^{2+}$ in bulk solution, considering the high binding constants (~$5 \times 10^6 M^{-1}$) and the large excess of G-quadruplex (2:1 of [HT]/[Ru]). The emission titration experiments (see Supplementary methods for Emission spectra titrations, Supplementary Fig. 4) confirm the complete binding of Ru complex at this ratio, as manifested by the saturated luminescence intensity at the [HT]/[Ru] ratio of 1:1–1:2. Therefore, the opening of the

the TTTT loop and the terminal quartet are coordinated to water molecules. A number of water molecules link the phosphate groups of thymine residues and bridge T bases with $Na^+$ ion, which are the key interactions to maintain the integrity of loop conformation and the structural integrity of G-quadruplex. Therefore, the bridging water molecules existent in TTTT loops, although not mobile waters, can form hydrogen bond with the dppz ligand and thus lead to the opening of the dark state relaxation channel for $[Ru(phen)_2dppz]^{2+}$ bound to the *Oxyticha nova* G-quadruplex. The mobility of these water should be less than the dynamic waters entering the flexible bilateral TTA loops in HT G-quadruplex, which might be the reason accounting for the slower dark state formation rate in the *Oxyticha nova* G-quadruplex.

These observations indicate further that the variations of the microenvironment hydration pattern of G-quadruplex DNA can be sensitively probed by the ultrafast photodynamics of $[Ru(phen)_2dppz]^{2+}$. Intrinsically, this sensitivity is related to the energetics of the bright $^3$MLCT state and the dark state (Fig. 1)[2,46,48]. There is a dynamic equilibrium between the bright state and the lower-lying dark state, and the dark state is favoured enthalpically ($\Delta H° < 0$) in water[2,48]. Water will stabilize the dark state relative to the bright state by hydrogen bonds with phenazine nitrogen atoms. The energy gap between the dark and bright states becomes larger (i.e., more negative $\Delta H°$), which will make the equilibrium inclined to the dark state and thus the dark state formation can be facilitated in water. In aprotic solvent such as $CH_3CN$, the energy of the dark state is less stabilized (i.e., less negative $\Delta H°$) and much less favoured, so the bright state population is dominant[10]. In the local bound DNA microenvironments, the relative population of the bright and dark states is thus highly associated with the microenvironment hydration patterns of DNA. In ds-DNA, the base-pair region is mostly hydrophobic as in $CH_3CN$, so the dark state channel is totally inhibited and bright state emission is dominant[45]. In HT G-quadruplex, the quartets and loops offer protection to allow light-switch, but there are dynamic waters entering the bilateral TTA loops where $[Ru(phen)_2dppz]^{2+}$ binds, providing a partially hydrophilic microenvironment that can drive the equilibrium to the dark state formation to some extent. Similarly, the dark state formation is observed in *Oxyticha nova* G-quadruplex, but with slower rate, probably because of the less amount of and less mobile bridging waters in the TTTT diagonal loop where $[Ru(phen)_2dppz]^{2+}$ binds. Accordingly, the quench of the dark state to ground state also becomes slower (517 ps) in *Oxyticha nova* than in HT G-quadruplex (482 ps), since this decay process is also highly associated with the polarity and proton-donating ability of the local solvent in the microenvironment[10], where the hydrogen bonds with the phenazine nitrogens are accepting modes for the radiationless decay. It should be also noted that there is only partial population branching to the dark state, and the long-lived bright states will still be primarily populated due to the partially hydrophobic character of the local bound microenvironment in G quadruplex. Thus, there is only partial (modest) ground state recovery being observed. On the 1000 ps timescale, only the dark state relaxation to the ground state accounts for the ground state recovery.

The ultrafast photodynamics results here indicate that the microenvironment hydration patterns of G-quadruplex can greatly influence the local excited state dynamics of the bound ruthenium complex, which has important implications for understanding the applications of Ru (II) complex as cellular photoprobes in recognizing different DNA structures. Particularly, given that the G-quadruplex DNA is abundant in telomere of cancerous cells[49], the importance of ruthenium complex as potential photoprobes to discriminate cancerous cell can be

underlined further. Intriguingly, when the dinuclear Ru (II) complex $[(phen)_2Ru(tpphz)Ru(phen)_2]^{4+}$ was applied *in cellulo* DNA stain, the feature of the bright state emission on binding with G-quadruplex (maxima of ~630 nm) different from ds-DNA (>650 nm) was observed[14], which suggested that this complex might be a promising DNA structural probe, specifically for G-quadruplex. Our results here reveal the clearly different ultrafast photodynamical relaxation pathways for Ru(II) complex in G-quadruplex compared with that in ds-DNA, which are of vital importance in terms of discerning G-quadruplex structure from ds-DNA. Particularly, it shows here that the ultrafast dark state channel tends to be more sensitive to the variation of the local DNA microenvironment (structure) than the bright state emission decay pathways, indicating that ultrafast pump-probe spectroscopic methods should be valuable in applications of signalling or imaging biological cells with abundance of certain DNA secondary structures[50].

## Conclusions

In summary, the light switch luminescence and ultrafast photo dynamics of $[Ru(phen)_2dppz]^{2+}$ in G-quadruplex DNA have been thoroughly investigated by multiscale time-resolved spectroscopy methods. $[Ru(phen)_2dppz]^{2+}$ binds strongly with both HT and *Oxyticha nova* G-quadruplex, showing pronounced light-switch effect. The nanosecond luminescence lifetime studies reveal a biexponential decay dynamics for the $^3$MLCT emission at 620 nm. Together with the mismatch control experiments, the binding modes (stacking of dppz ligand on the terminal G-quartet fully and partially) are disclosed explicitly. Further, femtosecond transient absorption spectroscopy measurements were performed for $[Ru(phen)_2(dppz)]^{2+}$ in G-quadruplex compared with in bulk water and in ds-DNA. By carefully inspecting the different spectral pattern and temporal evolution, the ultrafast dark state channel after fs 400 nm laser excitation has been discerned from the long-lived $^3$MLCT offset for $[Ru(phen)_2dppz]^{2+}$ in G-quadruplexes. It is found that the inhibited dark state channel in ds-DNA is open in G-quadruplex, but with a slower rate of formation compared to that in water, indicating that the local bound microenvironment in G-quadruplex is partially hydrophilic and partially hydrophobic.

The ultrafast dark state formation and decay rates are determined and found to be sensitive to the content of water molecules in local G-quadruplex structures. For $[Ru(phen)_2dppz]^{2+}$ bound to the bilateral TTA loops of HT, dark state formation rate is ~10.5 ps, which is close to that in 1:2 $CH_3CN/H_2O$ (10 ps). This reflects the more hydrophilic microenvironment of bilateral TTA loop region, where the flexible opening can allow dynamic entering of water molecules. For $[Ru(phen)_2dppz]^{2+}$ bound to the diagonal TTTT loops of *Oxyticha nova*, dark state formation rate is slower (~15.6 ps), which consists with the less amount of and less mobile bridging waters existent in the TTTT diagonal loop. The unique local excited state dynamics of $[Ru(phen)_2(dppz)]^{2+}$ in G-quadruplex is allowed to be characterized by ultrafast time-resolved spectroscopy, which provides mechanistic foundation for developing ruthenium complex as sensitive ultrafast photoprobes targeting G-quadruplex DNA structures and other related biological applications. We hope this work could arouse future research interests addressing photoactive ruthenium complex interactions with noncanonical DNA motifs such as G-quadruplex, which should be a topic with growing importance given the recent discovery of such structures in human cells[51,52].

## Methods

$[Ru(phen)_2dppz](BF_4)_2$ (phen = 1,10-phenanthroline; dppz = dipyridyl[3,2-a:2′,3′-c] phenazine) were prepared by the reported procedure and the concentration was determined using the known extinction coefficient of $2.23 \times 10^4 \, M^{-1} \, cm^{-1}$ at 439

nm[36]. The Human telomeric DNA oligonucleotides d[AG$_3$(T$_2$AG$_3$)$_3$] and *Oxyticha nova* telomere d(G$_4$T$_4$G$_4$) were purchased from the Sangon Biotech (Shanghai) Co., Ltd. in the ULTRAPAGE-purified form. The absorbance in the UV–vis spectra was monitored at 260 nm to determine single-strand concentrations based on the corresponding extinction coefficients of $2.285 \times 10^5$ and $1.152 \times 10^5$ M$^{-1}$ cm$^{-1}$[18]. The oligonucleotides were dissolved in a buffer solution containing 10 mM Tris–HCl, and 100 mM NaCl for at pH 7.5. The mixture was then first heated to 95 °C for 5 min before it was cooled down to room temperature with a cooling rate of 0.5 °C/min, and then incubated at 4 °C for 12 h. Meanwhile Human telomeric DNA oligonucleotides 5′-d[AG$_3$(T$_2$AG$_3$)$_3$]-3′ and its complementary sequence 5′-d[(C$_3$TA$_2$)$_3$C$_3$T]-3′ were used for forming the double-stranded DNA (ds-DNA) by the same procedure. The CD spectra were then measured to prove the formation of the targeted DNA structures.

Nanosecond time-resolved laser flash photolysis (LFP) was used to measure the $^3$MLCT luminescence decay dynamics[53]. Briefly, the instrument comprises an Edinburgh LP920 spectrometer (Edinburgh Instrument Ltd.) combined with an Nd:YAG laser (Surelite,Continuum Inc.). The excitation wavelength is 355 nm (1 Hz, fwhm ≈ 7 ns, 10 mJ/pulse). A monochromator equipped with a photomultiplier was used to collect the spectra with a range from 350 to 600 nm. The probe and pump beams are perpendicular and overlapped into the deoxygenated sample in a $10 \times 10$ mm cell. The data were transferred to a personal computer after the signals from the photomultiplier were displayed and recorded as a function of time on a 100 MHz (1.25 Gs/s sampling rate) oscilloscope (Tektronix, TDS 3012B). Data were analyzed by the online software of the LP920 spectrophotometer.

The femtosecond transient absorption spectra were measured at ~100 fs time resolution using a home-built femtosecond pump–probe setup, which has been described in detail elsewhere[54]. Briefly, a regeneratively amplified Ti:sapphire laser (Coherent Legend Elite) produces 40 fs, 1 mJ pulses at a 500 Hz repetition rate at 800 nm with a bandwidth (FWHM) of ~30 nm. The output from the amplifier is split by a 90/10 beam splitter to generate pump and probe beams. A portion of the 800 nm pulse was doubled with a 0.5 mm-thick BBO (type I) crystal to provide the 400 nm pump pulse or was used to generate 350 nm through optical parametric amplifiers (TOPAS, Light Conversion). The probe beam at 800 nm was sent to a computer-controlled optical delay line and then focused onto a 2 mm-thick water cell to generate a white light continuum that was split into two beams using a broadband 50/50 beam splitter as the reference and signal beams. The focused signal and pump beams were overlapped into a cuvette with a 1 mm beam path length, and the reference beam passed through the unexcited part of the sample. The cuvette was stirred during the data collection process to avoid thermal effects. The polarizations of pump and probe beams were set to 54.7° in all measurements. The OD at 400 nm of [Ru(phen)$_2$dppz]$^{2+}$ was kept at ~0.1. The concentration of [Ru(phen)$_2$dppz]$^{2+}$ was 66 μM, and the concentration of DNA was 132 μM (per quadruplex unit). The differential absorbance $\Delta A(t,\lambda)$ obtained by the femtosecond transient absorption spectra as a function of wavelength and time delay was corrected for the frequency-chirp and then analysed using the population dynamics modelling toolbox software Glotaran developed by van Wilderen et al.[55].

## Data availability
The authors declare that the data supporting the findings of this study are available within the paper and its supplementary information files.

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

## Acknowledgements
This work was financially supported by the National Natural Science Foundation of China (Grant Nos. 21933005, 21727803, 21425313, and 21703011).

## Author contributions
C.F.Y. and H.M.S. wrote the manuscript; H.M.S. designed the research. C.-H.H. and B.Z.Z. provided the ruthenium complex. C.F.Y., Q.Z., and Z.Q.J. H.M.Z. performed the experiments. All authors discussed the results and reviewed the manuscript.

## Competing interests
The authors declare no competing interests.
