## [Peer Review File · Communications Chemistry]

Reviewers' comments:

Reviewer #1 (Remarks to the Author):

Recommendation: This paper is publishable subject to minor revisions noted. Further review is not needed.

Comments:

This manuscript reports investigations on the photophysical processes of $[\text{Ru}(\text{phen})_2(\text{dppz})]^{2+}$ in the presence of several G-quadruplexes, using sub-picosecond transient absorption and nanosecond-resolved emission techniques. It was determined using circular dichroism melting points and lifetime measurements of G-T substituted DNA sequences that metal complexes are primarily bound to G-quadruplexes via end-stacking, close to the G8-position of the HT sequence. This binding position is different from the fully intercalated states of dual-strand DNA, and results in more dynamic relaxation behavior. They propose that these dynamics stem from semi-localized waters that aid in the structural stability of the solvated G-quadruplexes.

These findings are important for expanding the ideas of DNA-binding metal complexes to broader systems, which has been a lofty research goal for years. The results are of good quality, and the story-line is rational. However, as is the case with particularly interesting topics, it brought about several questions/suggestions, which require minor revisions to address:

1. I am curious about the accuracy of the early-time transient absorption spectra presented. Are these spectra corrected for the frequency-chirp of the probe pulse? This should be stated in the methods.

Additionally, the uncertainties in global fit parameters they list in Table 2 seem more accurate than the data presented in Figure 4 could achieve. For example, the time scale of 2.4 ± 0.03 ps seen for the metal complex in water would have decayed to essentially background (1.6% of the original signal) at 10 ps, which is the second-shortest time presented. I am curious if there was more data collected that is not presented in Figure 4 such that authors could make claims of such high accuracy. I don't think that more data/description needs to be presented in the paper as it may clutter the figure, but am simply curious about how the presented lifetimes were determined.

2. The authors briefly state that there is little enantioselectivity of the metal complex with ds-DNA, seeming to imply that this is also the case for the G-quadruplexes. However, enantioselectivity has been observed for similar metal complexes with G-quadruplexes using steady-state emission. {Hu, X. et. al., Dalton Trans. 2018, 47, 5422-5430} Given the more complex excited state dynamics seen, this may merit more of a discussion.

On that note, I am not completely convinced by the model the authors ascribe to the transient absorption lifetimes of metal complexes with G-quadruplexes; in particular, the shortest 2-3 ps lifetime ascribed to inter-system crossing, solvent reorganization and vibrational cooling. I would be quite surprised to hear that inter-system crossing occurs on ps timescales, as this has been observed in several Ruthenium complexes to be on the order of 15-40 fs. {Dongare P. et al. Coord. Chem. Rev. 2017, 345, 86-107} This lifetime is faster than many solvent reorganization times, and so should be essentially constant in different environments unless the ground state is considerably perturbed. Observations of vibrational cooling also seem unlikely for the phenanthroline ligands, which have been shown to have negligible cooling dynamics observed in the visible spectrum, in contrast to bipyridine counterparts. {Stark et al. J. Phys. Chem. A 2015, 119, 4813-4823} I am not sure if there

has been a study isolating vibrational cooling processes of the dppz ligand (e.g. in acetonitrile), although I would expect the the dppz ligand also shows minor spectral changes, due to its similar structure to phenanthroline. Solvent reorganization could possibly explain the slower kinetics, but these would be very dependent on the precise local environment.

The similarity of this short lifetime to that of bulk water (2-3 ps) suggests that perhaps in some bound complexes, the dark state is accessible on the same timescale as water, but are prohibited from relaxing further due to the local environment. This may stem from different specific binding sites or enantiomer orientations, but the result is that G-quadruplexes could have several different "types" of hydration at these positions resulting in two distinct lifetimes. This is rather rough speculation, but I am curious if authors have reasons to discredit such a picture of multiple binding orientations/dynamics.

3. The authors mention that the binding constant observed for G-quadruplex is similar to that for ds-DNA, but I was a little surprised that they did not compare the observed binding constant with other complexes that bind G-quadruplexes. {Shi S. et al. *Biochimie* 2010, 92, 370-377}

4. Keeping with the terminology used in the supplementary equation, I believe the apparent extinction coefficient (ϵ_a) should be labeled (A/Ct) , rather than $(A/[M])$.

5. The sentence on line 221-222 discusses inter-system crossing, solvent reorganization and vibrational cooling of Ruthenium complexes, yet the reference (18) is an article on circular dichroism of DNA Oligonucleotides. I suggest authors recheck other possible reference errors.

Reviewer #2 (Remarks to the Author):

In this communication, Su et al. reports on the dynamics of interaction of the well known $Ru(phen)_2(dppz)_2$ light-switch with G-quadruplexes. In order to get insight on the more precise localization of the complex within the quadruplex, they chose an elegant strategy by preparing and examining mismatched G-quadruplexes, which allow drawing interesting conclusions, and explored dynamical processes at very short timescales. This study is very interesting and deserves publication. However, some data are missing and some important points require clarification and modification prior to publication.

Steady state and nanosecond time-resolved spectroscopy

The authors mention that the Ru complex mainly stacks on the terminal of the G8-quartet. They show that, with HT10 mismatch, the lifetimes (especially the short one) are also affected, which indicates that the complex also binds this site. Therefore, why do the authors have bi-exponential fitting and thus only two lifetimes, whereas 4 different environments are present and occupied? The authors should discuss this point in their manuscript.

It would be also interesting to discuss in the manuscript the preexponential factors which strongly differ with the considered G-quadruplexes.

In table 1, data concerning the interaction in the presence of Oxyticha nova G-quadruplex are missing while they should help the discussion concerning the different types of sites later in the manuscript. The authors should add these data and discuss them also.

Ultrafast photodynamics

“*in the transient absorption spectra (Fig. 4a) the excited state absorption (ESA) bands of ³MLCT state peaking at 600 nm is observed initially, with its decay in the first 1 ps, there is a wavelength shift and build-up of the dark state band at 550 nm, which then fully decays to baseline within 1000 ps.*”

This looks somewhat difficult to combine with data reported in literature (and the authors cited these references) which show that the dark state in water is lower in energy than the first one populated at very short time scale. This is also mentioned by the authors themselves later in their manuscript (*"There is a dynamic equilibrium between the bright state and the lower-lying dark state, and the dark state is favoured enthalpically ($\Delta H^\circ < 0$) in water."*). How can we rationalize that the population of this lower-lying dark state is higher in energy ? The authors should comment on their data (reproducibility, error,...) and discuss that point in the manuscript.

“*Interestingly, the fs transient absorption spectra for [Ru(phen)₂dppz]²⁺ in HT G-quadruplex is clearly different from that in ds-DNA. As shown in Fig. 4c, the temporal evolution of the fs TA spectra displays a partial recovery of ground state bleach (440 nm), in concomitant with the decay of flat absorption ~ 700 nm and the discernible build-up around 600 nm.*”

When looking at figure 4(b) and 4(c), this is not clear and not convincing, especially for the build-up at 600 nm and also due to the fact that spectra in presence of ds-DNA are noisy (what is the reason for that ?). The authors should change this part and refer only to their data summarized in table 2. To give all the data allowing the discussion, they should give the information from global analysis in the presence of ds-DNA also, both in table 2 and in figure 5.

The ultrafast transient absorption spectra upon 350 nm excitation should be given in SI for the four systems.

The authors mention that the dark state channel is open when the Ru-complex is bound to the G-quadruplexes, and not open with ds-DNA, and that the protections are different depending on the quadruplex. Steady state emission intensities should help this discussion as different and opposite effects are at work. Much later in the manuscript (and it should be advantageously mentioned already earlier in the section dedicated to the ultrafast photodynamics), they mention *"It should be also noted that there is only partial population branching to the dark state, and the long-lived bright states will still be primarily populated due to the partially hydrophobic character of the local bound microenvironment in G quadruplex."*

This also should be discussed in link with emission intensities in the presence of the natural and mismatch G-quadruplexes.

Other comments.

Figure 1: buffer, salt and pH conditions should be added in the legend (and not only in material and methods) as they strongly influence structuration and stability of the G-quadruplex

“As a result, the replacement of T base (mismatch) tends to weaken the π - π stacking of G-quartets³² and such changes should be helpful for the ruthenium complexes penetrating deeper into the G-quadruplex core, it is expected that the emission lifetime of [Ru(phen)₂dppz]²⁺ should be increased if the mismatch point is the binding site.”

“penetrating deeper into the G-quadruplex core” is misleading and could suggest intercalation within the stacking of G4, which is not what is meant. This could be modified.

Reviewer #3 (Remarks to the Author):

Yang et al. utilized [Ru(phen)₂dppz]²⁺ as environmentally sensitive photoprobes for visualization of the microenvironment hydration properties of G-quadruplexes and to probe the local hydration effect on excited state dynamics. They showed that the microenvironment hydration patterns of G-quadruplex can greatly influence the local excited state dynamics of the bound ruthenium complex, which has important implications for understanding the applications of Ru (II) complex as cellular photoprobes in recognizing different DNA structures. The result is interesting and meaningful. The manuscript is well-written and could be published after addressing my concerns.

- 1.As we known, the phosphorescence intensity is strongly dependent on the oxygen concentration in solution. The phosphorescence lifetime and transient absorption spectra were conducted in degassed solution or air-saturated solution?
- 2.The phosphorescence lifetime of the 3MLCT of [Ru(phen)₂dppz]²⁺ in other solvents that cannot evolve into the dark state should be given, and it is easy for the reader to compare with the lifetime in water or binding with G-quadruplex.
- 3.How was the structure in Figure 3 obtained?
- 4.In figure 4a, the absorption peak has an obvious redshift when the dark state was formed from the 3MLCT in water. However, this redshift is difficult to be distinguished when the dark state was formed in HT G-quadruplex. Please give the reason.
- 5.To verify the accuracy of the global fitting, the comparison between the raw data and the fitting data should be provided.

Reviewer #1 (Remarks to the Author):

Recommendation: This paper is publishable subject to minor revisions noted. Further review is not needed.

Comments:

This manuscript reports investigations on the photophysical processes of $[\text{Ru}(\text{phen})_2(\text{dppz})]^{2+}$ in the presence of several G-quadruplexes, using sub-picosecond transient absorption and nanosecond-resolved emission techniques. It was determined using circular dichroism melting points and lifetime measurements of G-T substituted DNA sequences that metal complexes are primarily bound to G-quadruplexes via end-stacking, close to the G8-position of the HT sequence. This binding position is different from the fully intercalated states of dual-strand DNA, and results in more dynamic relaxation behavior. They propose that these dynamics stem from semi-localized waters that aid in the structural stability of the solvated G-quadruplexes.

These findings are important for expanding the ideas of DNA-binding metal complexes to broader systems, which has been a lofty research goal for years. The results are of good quality, and the story-line is rational. However, as is the case with particularly interesting topics, it brought about several questions/suggestions, which require minor revisions to address:

1. I am curious about the accuracy of the early-time transient absorption spectra presented. Are these spectra corrected for the frequency-chirp of the probe pulse? This should be stated in the methods.

Reply:

These spectra had been corrected for the frequency-chirped by the software Glotaran. We added this information in page 9. "The differential absorbance $\Delta A(t, \lambda)$ obtained by the femtosecond transient absorption spectra as a function of wavelength and time delay was corrected for the frequency-chirp and then analysed using the population dynamics modelling toolbox software Glotaran developed by van Wilderen et al."

Additionally, the uncertainties in global fit parameters they list in Table 2 seem more accurate than the data presented in Figure 4 could achieve. For example, the time scale of 2.4 ± 0.03 ps seen for the metal complex in water would have decayed to essentially background (1.6% of the original signal) at 10 ps, which is the second-shortest time presented. I am curious if there was more data collected that is not presented in Figure 4 such that authors could make claims of such high accuracy. I don't think that more data/description needs to be presented in the paper as it may clutter the figure, but am simply curious about how the presented lifetimes were determined.

Reply:

The lifetimes were obtained by the global fitting and the error bar originated from the fitting uncertainties. We illustrated this in the Table 2 legend. The fitting uncertainties can be very

small. Because of the good S/N ratio for the fs-TA spectra of the Ru complex in water, the fitting uncertainties is quite small (2.4 ± 0.03 ps). In contrast, the fitting uncertainties for the Ru complex in G-quadruplex are larger (2.7 ± 0.1 ps), due to the weaker signals of the fs-TA spectra (Hypochromic effect is usually seen in the absorption spectra of a ligand upon binding to DNA.).

The authors briefly state that there is little enantioselectivity of the metal complex with ds-DNA, seeming to imply that this is also the case for the G-quadruplexes. However, enantioselectivity has been observed for similar metal complexes with G-quadruplexes using steady-state emission. {Hu, X. et. al., Dalton Trans. 2018, 47, 5422-5430} Given the more complex excited state dynamics seen, this may merit more of a discussion.

Reply:

Thanks for the valuable suggestions.

The previous studies {J. Am. Chem. Soc. 1993, 115, 3448-3454} showed that both Λ and Δ enantiomers of $[\text{Ru}(\text{phen})_2(\text{dppz})]^{2+}$ have similar strong binding constants to ds-DNA, indicating little enantioselectivity. The Δ enantiomer has stronger emissions upon ds-DNA binding, and the reason was ascribed to a slightly different intercalation geometry causing a different degree of protection from quenching by water, instead of enantioselectivity.

In fact, the situation for G-quadruplex is very similar to ds-DNA. Both Λ and Δ enantiomers have strong binding constants to the G-quadruplex. The binding constants of the analogue $[\text{Ru}(\text{phen})_2\text{dppz}]^{2+}$ with HT G-quadruplex are $2.7 \times 10^6 \text{ M}^{-1}$ for Δ and $4.6 \times 10^6 \text{ M}^{-1}$ for Λ enantiomers, respectively. {Chem. Eur. J. 2015, 21, 11435-11445 } Hu showed that Λ enantiomer has stronger emission upon binding with HT G-quadruplex. {Hu, X. et. al., Dalton Trans. 2018, 47, 5422-5430}

Thus, in analogy to ds-DNA, there is negligible enantioselectivity of $[\text{Ru}(\text{phen})_2\text{dppz}]^{2+}$ binding with HT G-quadruplex, although differential emission intensities were observed between the Δ and Λ enantiomers with G-quadruplex, suggesting slightly different binding geometries for each enantiomer. We add brief discussions in page 4.

On that note, I am not completely convinced by the model the authors ascribe to the transient absorption lifetimes of metal complexes with G-quadruplexes; in particular, the shortest 2-3 ps lifetime ascribed to inter-system crossing, solvent reorganization and vibrational cooling. I would be quite surprised to hear that inter-system crossing occurs on ps timescales, as this has been observed in several Ruthenium complexes to be on the order of 15-40 fs. {Dongare P. et al. Coord. Chem. Rev. 2017, 345, 86-107} This lifetime is faster than many solvent reorganization times, and so should be essentially constant in different environments unless the ground state is considerably perturbed. Observations of vibrational cooling also seem unlikely for the phenanthroline ligands, which have been shown to have negligible cooling dynamics observed in the visible spectrum, in contrast to bipyridine counterparts. {Stark et al. J. Phys. Chem. A 2015, 119, 4813-4823} I am

not sure if there has been a study isolating vibrational cooling processes of the dppz ligand (e.g. in acetonitrile), although I would expect the the dppz ligand also shows minor spectral changes, due to its similar structure to phenanthroline. Solvent reorganization could possibly explain the slower kinetics, but these would be very dependent on the precise local environment.

The similarity of this short lifetime to that of bulk water (2-3 ps) suggests that perhaps in some bound complexes, the dark state is accessible on the same timescale as water but are prohibited from relaxing further due to the local environment. This may stem from different specific binding sites or enantiomer orientations, but the result is that G-quadruplexes could have several different “types” of hydration at these positions resulting in two distinct lifetimes. This is rather rough speculation, but I am curious if authors have reasons to discredit such a picture of multiple binding orientations/dynamics.

Reply:

We agree that the intersystem crossing to form $^3\text{MLCT}$ should occur on fs time scale in this ruthenium complex. In G-quadruplex due to the restriction of DNA, the solvation of $^3\text{MLCT}$ becomes much slower than that in water. So the lifetime of 2-3 ps should be ascribed to the combination of the ISC to $^3\text{MLCT}$ and the solvation of $^3\text{MLCT}$. This assignment is consistent with reports for related Ru complex {*J. AM. CHEM. SOC.* 2010, *132*, 5594–5595}. To make it clearer, we removed “vibrational cooling” and changed the words in page 6 “The shortest component ($\tau_1 = 2.7$ ps) is attributed to a combination of intersystem crossing (should be as fast as in water) and solvent reorganization of $^3\text{MLCT}$, consistent with reports for related Ru complexes.⁴⁵ This time constant should mainly reflect the solvent reorganization process, which is usually slower than that in water because of DNA structure restriction.”

We do not agree with the reviewer’s speculation “The similarity of this short lifetime to that of bulk water (2-3 ps) suggests that perhaps in some bound complexes, the dark state is accessible on the same timescale as water but are prohibited from relaxing further due to the local environment.....”. We didn’t attribute this time constant to the dark state formation in G-quadruplex: 1)the dark state formation must come from $^3\text{MLCT}$, so it is reasonable to attribute this first process (2-3 ps) to the formation of $^3\text{MLCT}$ instead of the dark state; 2) if there is a local environment in G-quadruplex that makes the dark state to form in 2-3 ps behaving like in water, the dark state should also decay in 248 ps, but was not observed in our experiments. Actually, the decay of the dark state in G-quadruplex is obvious much slower (~482 ps). ; 3) The dark state decay process is also highly associated with the polarity and proton-donating ability of the local solvent in the microenvironment {*J. Am. Chem. Soc.* 1997, **119**, 11458-11467}. If the local environment slows down the dark state formation, the dark state quench to ground state should also become slower. Therefore, it is reasonable to assign τ_2 (10.5 ps) to the $^3\text{MLCT}$ state relaxation forming the dark state and τ_3 (482 ps) to the dark state quench to ground state in G-quadruplex, which are both slower than in water.

3. The authors mention that the binding constant observed for G-quadruplex is similar to that for ds-DNA, but I was a little surprised that they did not compare the observed binding constant with other complexes that bind G-quadruplexes. {Shi S. et al. Biochimie 2010, 92, 370-377}

Reply:

Thanks for the advice.

We added the comparison and the related reference in Page 2: "This binding constant is comparable to that for these complexes with ds-DNA ($\sim 10^6 \text{ M}^{-1}$), and similar to the binding constant between $[\text{Ru}(\text{bpy})_2\text{dppz}]^{2+}$ and G-quadruplex".

4. Keeping with the terminology used in the supplementary equation, I believe the apparent extinction coefficient (ea) should be labeled (A/Ct), rather than (A/[M]).

Reply:

It has been changed to (A/ C_t).

5. The sentence on line 221-222 discusses inter-system crossing, solvent reorganization and vibrational cooling of Ruthenium complexes, yet the reference (18) is an article on circular dichroism of DNA Oligonucleotides. I suggest authors recheck other possible reference errors.

Reply:

We revised it in page 6. The correct reference is "Sun, Y., Liu, Y. & Turro, C. Ultrafast Dynamics of the Low-Lying (MLCT)-M-3 States of $[\text{Ru}(\text{bpy})_2(\text{dppz})]^{2+}$. J. Am. Chem. Soc. 132, 5594-+ (2010)."

Reviewer #2 (Remarks to the Author):

In this communication, Su et al. reports on the dynamics of interaction of the well known $\text{Ru}(\text{phen})_2(\text{dppz})^{2+}$ light-switch with G-quadruplexes. In order to get insight on the more precise localization of the complex within the quadruplex, they chose an elegant strategy by preparing and examining mismatched G-quadruplexes, which allow drawing interesting conclusions, and explored dynamical processes at very short timescales. This study is very interesting and deserves publication. However, some data are missing and some important points require clarification and modification prior to publication.

Steady state and nanosecond time-resolved spectroscopy

The authors mention that the Ru complex mainly stacks on the terminal of the G8-quartet. They show that, with HT10 mismatch, the lifetimes (especially the short one) are also affected, which indicates that the complex also binds this site. Therefore, why do the authors have bi-exponential fitting and thus only two lifetimes, whereas 4 different environments are present and occupied ? The authors should discuss this point in their manuscript. It would be also interesting to discuss in

the manuscript the preexponential factors which strongly differ with the considered G-quadruplexes.

Reply:

Thanks for the valuable suggestions.

As shown in Figure 2b, the emission decay kinetics of $[\text{Ru}(\text{phen})_2\text{dppz}]^{2+}$ binding with HT10 is about the same as that with HT. The fitted time constants for HT10 were double-checked and corrected in table 1, showing that they are not obviously different from those for HT. In contrast, the time constants are dramatically changed for HT8, indicating that the ruthenium complexes mainly stack on the G8-quartet.

Although the substitution of G by T in the terminal quartets don't change the wild-type's antiparallel fold in Na^+ buffer, the whole structure of G-quadruplex might be slightly perturbed. Therefore, we think the small changes of lifetimes and preexponential factors with HT10 is caused by the whole G-quadruplex structure perturbation upon the G10 replacement by T, instead of the Ru complex's binding at this site.

We add discussions in Page 4 to address these points: "Since the ruthenium complexes mainly stacks on the G8-quartet, the replacement of G10 by T doesn't lead to obviously different time constants in HT10 compared to HT. The change of the preexponential factors in HT 10 should be related to the small perturbation of the whole G-quadruplex structure upon G10 mismatch. This small perturbation might be beneficial for the ruthenium complexes to approach into the core of the G-quadruplex, which is implied by the increased preexponential factor of the long lifetime τ_L ."

In table 1, data concerning the interaction in the presence of *Oxyticha nova* G-quadruplex are missing while they should help the discussion concerning the different types of sites later in the manuscript. The authors should add these data and discuss them also.

Reply:

We add the experimental data of the $^3\text{MLCT}$ luminescence decay kinetics for $[\text{Ru}(\text{phen})_2\text{dppz}]^{2+}$ when bound to *Oxyticha nova* G-quadruplex upon 355 nm excitation in SI (Supplemental Fig. 6). And the fitted time constants ($\tau_L=124$ ns and $\tau_S=23$ ns) are added in Table 1. Likewise, the two emission lifetimes correspond to full stacking and partial stacking onto the terminal G-quartet. We add discussions in SI (Page 4).

Ultrafast photodynamics

"in the transient absorption spectra (Fig. 4a) the excited state absorption (ESA) bands of $^3\text{MLCT}$ state peaking at 600 nm is observed initially, with its decay in the first 1 ps, there is a wavelength shift and build-up of the dark state band at 550 nm, which then fully decays to baseline within 1000 ps."

This looks somewhat difficult to combine with data reported in literature (and the authors cited these references) which show that the dark state in water is lower in energy than the first one populated at very short time scale. This is also

mentioned by the authors themselves later in their manuscript (*"There is a dynamic equilibrium between the bright state and the lower-lying dark state, and the dark state is favoured enthalpically ($\Delta H^\circ < 0$) in water."*). How can we rationalize that the population of this lower-lying dark state is higher in energy ? The authors should comment on their data (reproducibility, error, ...) and discuss that point in the manuscript.

Reply:

The fs transient-absorption spectroscopy measures the excited state absorption (ESA) bands. For the $^3\text{MLCT}$ state, it absorbs a 600 nm photon to populate higher excited state. For the dark state, due to its lower-lying energy, it requires to absorb higher energy photon to populate higher excited state, so the ESA band is at 550 nm. Please see the following scheme.

"Interestingly, the fs transient absorption spectra for $[\text{Ru}(\text{phen})_2\text{dppz}]^{2+}$ in HT G-quadruplex is clearly different from that in ds-DNA. As shown in Fig. 4c, the temporal evolution of the fs TA spectra displays a partial recovery of ground state bleach (440 nm), in concomitant with the decay of flat absorption ~ 700 nm and the discernible build-up around 600 nm."

When looking at figure 4(b) and 4(c), this is not clear and not convincing, especially for the build-up at 600 nm and also due to the fact that spectra in presence of ds-DNA are noisy (what is the reason for that ?). The authors should change this part and refer only to their data summarized in table 2. To give all the data allowing the discussion, they should give the information from global analysis in the presence of ds-DNA also, both in table 2 and in figure 5.

The ultrafast transient absorption spectra upon 350 nm excitation should be given in SI for the four systems.

Reply:

Hypochromic effect is usually seen in the absorption spectra (including fs-TA spectra) of a ligand upon binding to DNA. Moreover, the local microenvironment of G-quadruplex where Ru complex binds is only partially hydrophilic, which makes the lowering of dark state energy relative to $^3\text{MLCT}$ state less than that in bulk water (Scheme 1). Therefore, the ESA band of dark state does not change much compared to the ESA band of $^3\text{MLCT}$ state, both of which fall at around 600 nm. This explains why the temporal evolutions of the Ru complex in G-quadruplex in fs-TA spectra are not as obvious as that in bulk water. Still, the small spectral change (the decay of flat absorption ~ 700 nm and the discernible build-up around 600 nm) due to the dark state formation can be discerned. The description of the spectral change combines with kinetics data to show the excited state relaxation picture.

For the case in ds-DNA, there is no kinetics evolution at all in the fs-TA spectra. So there is no need to perform global kinetics analysis.

At the suggestion, we add the fs-TA spectra obtained upon 350 nm excitation in SI (Supplementary Fig.11).

The authors mention that the dark state channel is open when the Ru-complex is bound to the G-quadruplexes, and not open with ds-DNA, and that the protections are different depending on the quadruplex. Steady state emission intensities should help this discussion as different and opposite effects are at work. Much later in the manuscript (and it should be advantageously mentioned already earlier in the section dedicated to the ultrafast photodynamics), they mention *“It should be also noted that there is only partial population branching to the dark state, and the long-lived bright states will still be primarily populated due to the partially hydrophobic character of the local bound microenvironment in G quadruplex.”*

This also should be discussed in link with emission intensities in the presence of the natural and mismatch G-quadruplexes.

Reply:

Thanks for the valuable suggestions. We have added Supplementary Fig.7 and corresponding discussions in SI to compare the emission intensities of the Ru complex in different DNA systems:

“When $[\text{Ru}(\text{phen})_2\text{dppz}]^{2+}$ is bound to ds-DNA, HT and Oxyticha nova, respectively, the steady state emission intensities follow the order: ds-DNA > HT > Oxyticha nova, which is consistent with the the time-resolved luminescence decay kinetics data. In contrast to the intercalation binding mode in ds-DNA, the stacking binding mode with quartets in G-quadruplex may provide less protection for the Ru complex from water quenching. Therefore, the emission is the strongest when bound to ds-DNA. Moreover, the bilateral TTA loop region of HT and the TTTT diagonal loop region of Oxyticha nova provide different protections for the Ru complex, and the binding constants for these two G-quadruplex are different (Supplementary Table 1: HT > Oxyticha nova), so the emission intensities of the Ru complex in HT is larger than Oxyticha nova.

For the mismatch G-quadruplexes in comparison to natural HT as shown in Figure 2(b), the emission intensity of the Ru complex follows the order: HT8 > HT~HT10, consistent with the the time-resolved luminescence decay kinetics data (Table 1). This agree with our discussion in the paper: 1) replacement of T base (mismatch) tends to weaken the π - π stacking of G-quartets and such changes should be helpful for the ruthenium complexes approaching G-quadruplex core, and can render phenazine nitrogens relatively well shielded by the quartet and loops; 2) $[\text{Ru}(\text{phen})_2\text{dppz}]^{2+}$ mainly stacks on the terminal of the G8-quartet with the lateral loops. The replacement of G10 by T doesn't lead to obviously different emission intensity in HT10. So, the ruthenium bound to HT8 has the strongest emission intensity.”

Other comments.

Figure 1: buffer, salt and pH conditions should be added in the legend (and not only

in material and methods) as they strongly influence structuration and stability of the G-quadruplex

Reply:

We add these conditions in the legend as suggested.

“As a result, the replacement of T base (mismatch) tends to weaken the π - π stacking of G-quartets³² and such changes should be helpful for the ruthenium complexes penetrating deeper into the G-quadruplex core, it is expected that the emission lifetime of [Ru(phen)₂dppz]²⁺ should be increased if the mismatch point is the binding site.”

“penetrating deeper into the G-quadruplex core” is misleading and could suggest intercalation within the stacking of G4, which is not what is meant. This could be modified.

Reply:

Thanks for the suggestions.

We made revisions in Page 4: “such changes should be helpful for the ruthenium complexes approaching G-quadruplex core, which will offer better protection from the water”

Reviewer #3 (Remarks to the Author):

Yang et al. utilized [Ru(phen)₂dppz]²⁺ as environmentally sensitive photoprobes for visualization of the microenvironment hydration properties of G-quadruplexes and to probe the local hydration effect on excited state dynamics. They showed that the microenvironment hydration patterns of G-quadruplex can greatly influence the local excited state dynamics of the bound ruthenium complex, which has important implications for understanding the applications of Ru (II) complex as cellular photoprobes in recognizing different DNA structures. The result is interesting and meaningful. The manuscript is well-written and could be published after addressing my concerns.

1. As we known, the phosphorescence intensity is strongly dependent on the oxygen concentration in solution. The phosphorescence lifetime and transient absorption spectra were conducted in degassed solution or air-saturated solution?

Reply:

The phosphorescence lifetime measurement by nanosecond timescale experiments was carried in the degassed solutions to prevent the quenching by O₂. The ultrafast fs-transient absorption spectra were not conducted under degassed conditions, since the quenching of the ³MLCT state by O₂ is a bimolecular collision process occurring on nanosecond timescale, which does not affect the fs-ps processes.

2. The phosphorescence lifetime of the ³MLCT of [Ru(phen)₂dppz]²⁺ in other solvents that cannot evolve into the dark state should be given, and it is easy for the reader to compare with the lifetime in water or binding with G-quadruplex.

Reply:

The phosphorescence lifetime of the $^3\text{MLCT}$ of $[\text{Ru}(\text{phen})_2\text{dppz}]^{2+}$ in other nonaqueous solvents had been studied in the reference {Inorg.Chem 1997, 36,962-965}. For instance, in the degassed CH_3CN solutions, the phosphorescence lifetime of the $^3\text{MLCT}$ is 663 ns. We add discussion in Page 3. "The $^3\text{MLCT}$ luminescence lifetimes of $[\text{Ru}(\text{phen})_2\text{dppz}]^{2+}$ in DNA systems are both shorter than that in pure nonaqueous solvent, which was reported to be 663 ns in CH_3CN ."

3. How was the structure in Figure 3 obtained?

Reply:

The structure of G-quadruplex is obtained from the PDB databank, which reports the structure elucidated by NMR measurements (PDB 143D). The binding of $[\text{Ru}(\text{phen})_2\text{dppz}]^{2+}$ with the G-quadruplex were plotted schematically. This information has been added in the legend of Fig.3.

4. In figure 4a, the absorption peak has an obvious redshift when the dark state was formed from the $^3\text{MLCT}$ in water. However, this redshift is difficult to be distinguished when the dark state was formed in HT G-quadruplex. Please give the reason.

Reply:

Thanks for the valuable suggestions.

This phenomena in turn demonstrates the distinct effect of G-quadruplex local environment on the dark state formation dynamics.

Hypochromic effect is usually seen in the absorption spectra (including fs-TA spectra) of a ligand upon binding to DNA. Moreover, the local microenvironment of G-quadruplex where Ru complex binds is only partially hydrophilic, which makes the lowering of dark state energy relative to $^3\text{MLCT}$ state less than that in bulk water (Scheme 1). Therefore, the ESA band of dark state does not change much compared to the ESA band of $^3\text{MLCT}$ state, both of which fall at around 600 nm. This explains why the temporal evolutions of the Ru complex in G-quadruplex in fs-TA spectra are not as obvious as that in bulk water. Still, the small spectral change (the decay of flat absorption ~ 700 nm and the discernible build-up around 600 nm) due to the dark state formation can be discerned. We add these explanations in Page 7.

5. To verify the accuracy of the global fitting, the comparison between the raw data and the fitting data should be provided.

Reply:

Thanks for your advices. We have provided the comparisons between the raw data and the fitting data (Supplementary Fig.8-10) in SI.

REVIEWERS' COMMENTS:

Reviewer #1 (Remarks to the Author):

The changes the authors have made to the manuscript have adequately addressed all issues that I initially had, and needs no further change.

In regards to speculation as to the mechanism of the 2-3 ps component, I agree with their second/third points (which are similar) that if there were a site that allowed solvation dynamics similar to that of pure water, it would most likely also display a subsequent decay faster than what is observed.

Their first point I disagree with only in terms of semantics, as they state earlier that the initial intersystem crossing "formation" of the 3MLCT state occurs on fs time scales. I would agree that subsequent "evolution" of the 3MLCT state to a stable quasi-equilibrium (i.e. constant rates of electron hopping, vibrations, etc.) could occur on the 2-3 ps time scale.

Reviewer #2 (Remarks to the Author):

The study is very interesting and deserves publication. This revised manuscript could be accepted for publication after a few changes. More precisely, two points still require modifications in the manuscript:

Previous comment: In table 1, data concerning the interaction in the presence of Oxyticha nova G-quadruplex are missing while they should help the discussion concerning the different types of sites later in the manuscript. The authors should add these data and discuss them also.

The data in emission are now present in SI and the lifetime data are in Table 1 but a discussion should appear in the manuscript (not in the SI) and it should be an accurate discussion related to the structure of Oxyticha nova G4 and its difference with the other G4s, which is not the case. This should be modified prior to acceptance.

“Interestingly, the fs transient absorption spectra for [Ru(phen)₂dppz]²⁺ in HT G-quadruplex is clearly different from that in ds-DNA. As shown in Fig. 4c, the temporal evolution of the fs TA spectra displays a partial recovery of ground state bleach (440 nm), in concomitant with the decay of flat absorption ~ 700 nm and the discernible build-up around 600 nm.”

Previous comment: When looking at figure 4(b) and 4(c), this is not clear and not convincing, especially for the build-up at 600 nm and also due to the fact that spectra in presence of ds-DNA are noisy (what is the reason for that ?). The authors should change this part and refer only to their data summarized in table 2.

Reply by the authors:

Hypochromic effect is usually seen in the absorption spectra (including fs-TA spectra) of a ligand upon binding to DNA. Moreover, the local microenvironment of G-quadruplex where Ru complex binds is only partially hydrophilic, which makes the lowering of dark state energy relative to 3MLCT state less than that in bulk water (Scheme 1). Therefore, the ESA band of dark state does not change much compared to the ESA band of 3MLCT state, both of which fall at around 600 nm. This explains why the temporal evolutions of the Ru complex in G-quadruplex in fs-TA spectra are not as obvious as that in bulk water. Still, the small spectral change (the decay of flat absorption ~ 700 nm and the discernible build-up around 600 nm) due to the dark state formation can be discerned. The

description of the spectral change combines with kinetics data to show the excited state relaxation picture.

I agree, but this discussion must be in the manuscript, not only in a reply to referees.

Reviewer #3 (Remarks to the Author):

The authors have sufficiently addressed all review comments in this revision, and acceptance is recommended.

Reviewer #1 (Remarks to the Author):

The changes the authors have made to the manuscript have adequately addressed all issues that I initially had, and needs no further change.

In regards to speculation as to the mechanism of the 2-3 ps component, I agree with their second/third points (which are similar) that if there were a site that allowed solvation dynamics similar to that of pure water, it would most likely also display a subsequent decay faster than what is observed.

Their first point I disagree with only in terms of semantics, as they state earlier that the initial intersystem crossing "formation" of the ³MLCT state occurs on fs time scales. I would agree that subsequent "evolution" of the ³MLCT state to a stable quasi-equilibrium (i.e. constant rates of electron hopping, vibrations, etc.) could occur on the 2-3 ps time scale.

Reply:

We have changed the descriptions in Page 7 .

Given that the intersystem crossing of [Ru(phen)₂dppz]²⁺ from ¹MLCT to ³MLCT is ~ 300 fs, the first time constant ($\tau_1 = 2.7$ ps) should be attributed to subsequent "evolution" of the ³MLCT state to a stable quasi-equilibrium *via* solvent reorganization or vibrational cooling, consistent with reports for related Ru complexes

Reviewer #2 (Remarks to the Author):

The study is very interesting and deserves publication. This revised manuscript could be accepted for publication after a few changes. More precisely, two points still require modifications in the manuscript:

Previous comment: In table 1, data concerning the interaction in the presence of *Oxyticha nova* G-quadruplex are missing while they should help the discussion concerning the different types of sites later in the manuscript. The authors should add these data and discuss them also.

The data in emission are now present in SI and the lifetime data are in Table 1 but a discussion should appear in the manuscript (not in the SI) and it should be an accurate discussion related to the structure of *Oxyticha nova* G4 and its difference with the other G4s, which is not the case. This should be modified prior to acceptance.

Reply : Thanks for the suggestions.

We have added the discussions in Page8

Since the TTTT diagonal loop region of *Oxyticha nova* provides protections for the Ru complex different from the bilateral TTA loop region of HT G-quadruplex, and the binding constants for these two G-quadruplex are different, the emission lifetimes of $\text{Ru}(\text{phen})_2\text{dppz}]^{2+}$ with *Oxyticha nova* G-quadruplex are different from that of $\text{Ru}(\text{phen})_2\text{dppz}]^{2+}$ with HT G-quadruplex (Supplementary Figure 6 and Table 1), which is consistent with the steady state emission intensities of $[\text{Ru}(\text{phen})_2\text{dppz}]^{2+}$ (Supplementary Figure 7)

“ Interestingly, the fs transient absorption spectra for [Ru(phen)2dppz]2+ in HT G-quadruplex is clearly different from that in ds-DNA. As shown in Fig. 4c, the temporal evolution of the fs TA spectra displays a partial recovery of ground state bleach (440 nm), in concomitant with the decay of flat absorption ~ 700 nm and the discernible build-up around 600 nm.”

Previous comment: When looking at figure 4(b) and 4(c), this is not clear and not convincing, especially for the build-up at 600 nm and also due to the fact that spectra in presence of ds-DNA are noisy (what is the reason for that ?). The authors should change this part and refer only to their data summarized in table 2.

Reply by the authors:

Hypochromic effect is usually seen in the absorption spectra (including fs-TA spectra) of a ligand upon binding to DNA. Moreover, the local microenvironment of G-quadruplex where Ru complex binds is only partially hydrophilic, which makes the lowering of dark state energy relative to 3MLCT state less than that in bulk water (Scheme 1). Therefore, the ESA band of dark state does not change much compared to the ESA band of 3MLCT state, both of which fall at around 600 nm. This explains why the temporal evolutions of the Ru complex in G-quadruplex in fs-TA spectra are not as obvious as that in bulk water. Still, the small spectral change (the decay of flat absorption ~ 700 nm and the discernible build-up around 600 nm) due to the dark state formation can be discerned. The

description of the spectral change combines with kinetics data to show the excited state relaxation picture.

I agree, but this discussion must be in the manuscript, not only in a reply to referees.

Reply:

This discussion has been added in the manuscript, the second paragraph in Page 8.

Reviewer #3 (Remarks to the Author):

The authors have sufficiently addressed all review comments in this revision, and acceptance is recommended.